# Efficiency Improvement Strategies for Public Health Systems: Developing and Evaluating a Taxonomy in the Australian Healthcare System

**DOI:** 10.3390/healthcare11152177

**Published:** 2023-07-31

**Authors:** James Kenneth Walters, Anurag Sharma, Reema Harrison

**Affiliations:** 1NSW Ministry of Health, Level 9, 1 Reserve Rd, St Leonards, NSW 2065, Australia; 2School of Population Health, Faculty of Medicine, UNSW, Kensington Campus, Level 2, Samuels Building (F25), Kensington, NSW 2052, Australia; anurag.sharma@unsw.edu.au; 3Centre for Health Systems and Safety Research, Faculty of Medicine, Health and Human Sciences, Macquarie University, Level 6, 75 Talavera Road, Sydney, NSW 2109, Australia; reema.harrison@mq.edu.au

**Keywords:** public health, health management, efficiency

## Abstract

Introduction: As demand for healthcare continues to grow, public health systems are increasingly required to drive efficiency improvement (EI) to address public service funding challenges. Despite this requirement, evidence of EI strategies that have been successful applied at the whole-of-system level is limited. This study reports the development, implementation and evaluation of a novel taxonomy of EI strategies used in public health systems to inform systemwide EI models. Materials and methods: The public health system in New South Wales, Australia, operates a centralised EI model statewide and was the setting for this study. An audit of EI strategies implemented in the NSW Health system between July 2016 and June 2019 was used to identify all available EI strategies within the study timeframe. A content management approach was applied to audit the strategies, with each strategy coded to an EI focus area. Codes were clustered according to similarity, and category names were assigned to each cluster to form a preliminary taxonomy. Each category was defined and examples were provided. The resulting taxonomy was distributed and evaluated by user feedback survey and pre–post study to assess the impact on EI strategy distribution. Results: A total of 1127 EI strategies were identified and coded into 263 unique strategies, which were clustered into nine categories to form the taxonomy of EI strategies. Categories included the following: non-clinical contracts and supplies; avoided and preventable activity; clinical service delivery and patient outcomes; finance and operations; recruitment, vacancies and FTE; staffing models; leave management; staff engagement and development; premium staffing; and clinical contracts and supplies. Evaluative findings revealed a perceived reduction in the duplication of EI work, improved access to EI knowledge and improved engagement with EI processes when using the taxonomy. The taxonomy was also associated with wider use of EI strategies. Conclusions: Whole-of-system EI is an increasing requirement. Using a taxonomy to guide systemwide practice appears to be advanta-geous in reducing duplication and guiding practice, with implications for use in health systems internationally.

## 1. Introduction

Despite great focus on efficiency improvement (EI) in public health systems, there has been a lack of focus on establishing processes to support EI at a whole-of-system level internationally [1]. A general focus on EI at hospital and clinical-unit levels has led to siloed work to generate EI strategies, resulting in duplication of work to tackle similar challenges within organisations in the same public health system [2]. Efficiency is conceptualised in a variety of ways in theoretical models [1]. In contemporary health system paradigms, it is recognised as a central component of value-based care. In this characterisation, efficiency represents the outcomes achieved for resources invested [1,3]. This exists in balance with other core elements of value, with a requirement to uphold or maintain or improve clinical, quality and experience outcomes in balance while reducing the cost of health service delivery [1].

Knowledge in the context of organisations is defined as information which adds value, supports decision-making or is useful in delivering outcomes [4,5]. A knowledge audit is recognised as a fundamental step in the processes of managing organisational knowledge, with a wide range of various models, frameworks and techniques for undertaking such audits evident in the literature [4]. Common elements across audit approaches involve the identification of existing knowledge, gaps in knowledge and opportunities to apply knowledge to support business processes and objectives [5]. In the context of approaches to efficiency and performance management, this also allows an organisation to understand the extent of current EI knowledge in order to prevent unnecessary effort wasted in replicating knowledge already generated [6].

A taxonomy represents a solution for providing structure for reporting and sharing unstructured data elements by organising them into categories which are closely related according to their requirements, purpose and/or focus area [7,8]. This structure and shareability creates value for organisational knowledge by promoting access to distributed knowledge to enhance organisational performance [9]. Taxonomies are also associated with the facilitation of discussion on complex topics, assisting organisations and the people within them to build shared understanding [10]. The aim of this study is to address the research question “how can access to and utilisation of EI knowledge at a systemwide level can be improved through the development and implementation of a taxonomy of EI strategies”?

In the current environment of ongoing increases in demand for healthcare amidst ongoing economic uncertainty, the increasing requirement for public health systems to maximise service outcomes achieved for resource investment is likely to persist. A gap in the literature exists regarding the design and delivery of centralised EI models, with a review of the recent literature noting a lack of comprehensive frameworks for driving EI at the whole-of-system level [1]. This study adds new knowledge towards filling this gap by providing a taxonomy for the classification and structuring of EI knowledge, as well as providing a catalogue of public health efficiency improvement strategies. This study has relevance for public health systems internationally seeking to classify EI approaches, structure EI programs and identify EI opportunities [8].

## 2. Materials and Methods

### 2.1. Setting

This study is set in New South Wales (NSW), a large Australian state with its own public health system, NSW Health. This system operates a systemwide EI model coordinated by a central program management office. The presence of this centralised approach in NSW is the rationale for selecting this setting. Through this model, each NSW Health service is required to achieve a mandated EI target each financial year by delivering EI strategies, with the EI strategy plans held and monitored by the central office in a database. These strategies aim to reduce the cost of delivering services while also improving or maintaining quality and experience outcomes, supporting public health organisations to meet service demand within allocated budgets and maximise outcomes achieved for resources invested [3]. Common examples include reducing reliance on costly contingent labour, changing to more cost-effective products, reducing low-value activity and reducing waste [6,11]. Similar improvement opportunities are present in all public health systems; however, for this distributed knowledge to be utilised, it must first be synthesised, organised and shared [12,13].

### 2.2. Design

A taxonomy development process was undertaken following an established intuitive content management approach to audit and form clusters of similar EI strategies [14,15].

### 2.3. Data Sources

The EI strategy database held by the NSW Health central EI office as outlined in the setting for this study was the source of data, making this study an audit of explicit knowledge as the study material existed in electronic format as formally documented information [5]. As this dataset is not publicly available, ethics approval for this study was obtained through the University of New South Wales (approval number HC201599).

### 2.4. Eligibility Criteria

To be eligible for inclusion, EI strategies must have been undertaken within the most recent three Australian financial years and held in the central NSW Health EI database. Only strategies which proceeded to implementation were included in this study, but the performance of these strategies is beyond the scope of this study.

### 2.5. Audit Procedure

To undertake the audit, data were extracted from each eligible EI strategy in the database, including financial year, the NSW Health organisation in which the improvement initiative was implemented, objectives, key metrics and financial assumptions. For each EI strategy, a one-sentence summary of the specific improvement focus area was derived to serve as a code for further analysis in alignment with content management methodology [15]. The metrics and calculations used to measure EI strategy performance were also extracted.

### 2.6. Taxonomy Development Procedure

In order to develop the EI taxonomy, the corresponding author systematically examined all one-sentence summary codes developed during the audit, clustering codes together in categories according to common areas of improvement focus using a content management approach [8,15]. Where multiple examples of identical EI strategies were identified, the assigned code was only included once in the category. Rigour was ensured by documenting the code for each data element. All elements were reviewed at audit completion by the corresponding author, with each element checked for an assigned code to ensure no items were overlooked.

An inductive method was applied to guide the categorisation process, with the units of analysis being the codes created through the knowledge audit procedure [14]. For each cluster, an overarching category label was assigned which best fit all items within the category as well as how they were referred to in practice [14,15]. Each category was sufficiently specific to avoid overlapping [9]. A second empirical check of categories was conducted at the completion of the categorisation process to ensure all codes fit within the defined categories with no ambiguity or outliers requiring additional labels to be added [10,14]. A flow chart of the audit and taxonomy development procedure is shown in Figure 1 below.

### 2.7. EI Resource Document Development, Implementation and Evaluation

The taxonomy was used to create a resource document to serve as a user guide for NSW Health EI stakeholders for the purpose of summarising and distributing EI knowledge across NSW Health.

Using the structure provided by the taxonomy categories, the EI strategy codes and performance measurement metrics for each unique EI strategy were listed to create a resource document which summarised the collective body of EI strategic knowledge from the sample data. Due to the clustering approach that was taken for classification, with all EI strategies grouped together underneath their respective taxonomy labels with duplicates removed, the structure for this resource was readily apparent and was set out according to the categories constructed during taxonomy development [9]. The resulting EI resource listed all 263 unique EI strategies identified through the audit. This resource was distributed to key EI stakeholders across the range of organisations responsible for clinical service delivery within the NSW public health system. This resource is provided as Appendix A.

### 2.8. Taxonomy Evaluation Methodology

Design: Following two and a half years of EI resource use, in March 2022, the impact of the EI resource was evaluated via a mixed-methods approach synthesising qualitative and quantitative data.

## 3. Qualitative Evaluation

### 3.1. Design

A series of free-text questions were developed to test the EI knowledge sharing concepts identified in the literature and to test the extent to which the EI resource had improved access to EI knowledge.

### 3.2. Ethics Approval

Modification for the existing ethics approval for the study was obtained through the University of New South Wales (Approval number HC210599).

### 3.3. Sample

A purposive sample of the population of designated NSW Health EI key stakeholders was the study group, which numbered 26 at the time of evaluation. This sampling approach aligns with the key informant technique [16]. Participant identities and demographics were not within the scope of this study and were not collected. Participants were recruited by an email invitation which included consent to participate along with an invitation to complete an anonymous online survey. The research team were blind to participant identity.

### 3.4. Data Collection

Free-text data were collected using MS Forms and extracted in MS Excel format, which was used to aggregate the responses for each question along with total number of responses received.

### 3.5. Analysis

Responses were clustered together but separated into individual statements to identify any recurring concepts, with responses also explored for potential linkages between themes [17]. An established thematic analysis checklist was used to support robust and systematic analysis throughout the process [17].

## 4. Quantitative Evaluation

### 4.1. Design

Observational pre/post study design was employed, using the body of NSW Health organisations as the study cohort and incidence of EI categories as the measure of comparison [18,19].

### 4.2. Sample and Data Collection

The existing data extract used for the knowledge audit sample included all EI plans from the most recent financial year prior to EI resource implementation. As such, this subsample was used as the baseline for the study (July 2018–June 2019). At the conclusion of the intervention financial year (July 2019–June 2020), the process used to audit and code EI plans for the creation of the taxonomy was repeated for all improvement plans within that period to create the comparison sample.

### 4.3. Analysis

Audit data for the pre/post financial years were combined in Microsoft Excel, with NSW Health organisation, financial year and taxonomy category compared using pivot tables. Through the comparison of results before and after the resource document was made available in the categories in which each NSW Health organisation reported at least one improvement plan, this pre–post comparison sought to establish whether providing the resource was linked to an increase in the number of organisations reporting improvement plans in each taxonomy category. This study also compared overall improvement plan numbers in each category before and after the taxonomy implementation. The pre–post difference of means and *p*-values for pre–post analysis both at the NSW Health organisation level and for overall EI strategy numbers were tested for statistical significance.

## 5. Results

### 5.1. EI Taxonomy Characteristics

The taxonomy development process resulted in eight categories. Of the 1127 EI strategies audited in this study, a total of 263 unique approaches were identified. Nine categories were identified with between 9 and 59 unique approaches in each category. Taxonomy labels were listed alphabetically. The eight categories within the taxonomy fit within three broader domains: employee-related expenditure, clinical services and non-clinical business. The list of unique strategies is provided in the taxonomy user guide resource document attached as Appendix A. EI category definitions, the number of unique EI strategies in each category, the alignment of taxonomy categorisations within broader improvement domains and EI strategy examples within each category are shown below in Table 1.

Each of the eight classifications within the taxonomy are explored as follows.

### 5.2. Clinical Services Domain

#### 5.2.1. Clinical Contracts and Supplies

EI strategies within this classification relate to the procurement and use of goods, consumables and commercial arrangements which are directly involved in the delivery of healthcare. This classification covers approaches such as reducing waste, changing to more cost-effective products or services and renegotiating contracts. Only EI strategies which focused on items, technologies and logistics required in the processes of healthcare were included in this category. Opportunities to improve efficiency represented by better-value products and providers were found to be a common area of opportunity across all NSW Health organisations. Examples include improving stock management to reduce waste, changing to more cost-effective service providers, working toward targeted reduction in high-cost consumables use and consolidating stock ordering by introducing standardised product catalogues.

#### 5.2.2. Clinical Service Delivery and Patient Outcomes

The remainder of the EI strategies within the clinical services domain focus on the processes of providing clinical care. With patient experience and patient outcomes included within the concept of value in healthcare, EI strategies in this classification also address opportunities to achieve patient-focused improvements while also reducing cost. This classification is distinct from the products and consumables approaches included in the previous classification in that the included approaches focus on reducing service time, reducing wasted time, improving asset utilisation and reviewing service provider arrangements. Examples include reducing unwarranted diagnostic procedures, improving patient transport scheduling to reduce out-of-hours transport requirements, improving appointment scheduling to minimise missed appointments and replacing paper forms with digital forms.

### 5.3. Workforce Domain

#### 5.3.1. Premium Staffing

Due to the requirement for health services to maintain minimum staffing levels often at short notice, a range of options to deploy staff at short notice are available; however, these are more costly than deploying regular staff. EI strategies include working additional or overtime hours, creating temporary contracts with medical staff and using staffing agencies to provide contingent workers. While these options provide flexibility and assure service capacity, EI strategies in this classification aim to reduce reliance on these options. Examples include reviewing billing practices of external providers, replacing contingent staff with permanent staff, implementing central staffing pools and improving overtime governance processes.

#### 5.3.2. Leave Management

EI strategies in this classification focus on ensuring appropriate and effective use of paid absence entitlements. This includes reducing sick leave, reducing accrued leave balances which would otherwise be paid out and reducing costs associated with replacing staff on planned leave. Common elements of these EI strategies include monitoring leave balances, monitoring utilisation rates and planning staff replacement in advance. Examples include reducing sick leave by targeting performance management, reducing excess annual leave liability, enhancing leave planning to minimise coverage costs and avoiding backfilling positions for short-term vacancies.

#### 5.3.3. Staffing Models

Health services typically have flexibility in how they deploy a mix of staff types to safely meet the demand for clinical services. EI strategies in this classification aim to reduce the cost of staffing while ensuring services are delivered to an appropriate level of safety, quality and accessibility. Such EI strategies include improving rostering practices, maximising use of role-delineated skillsets and aligning shift hours with service demand. Examples include increasing staffing hours to reduce after-hours callbacks, restructuring services to maximise skill mix utilisation, changing skill mix and seniority needed and improving governance to align staffing levels with activity.

#### 5.3.4. Recruitment, Vacancies and Full-Time Equivalent (FTE)

The remaining EI strategies in this domain relate to the establishment and filling of positions. While previously discussed contingent workforce options exist to meet short-term staffing pressures, most routine service staffing is achieved through the planned hiring and deployment of regular staff. This is measured in full-time equivalent (FTE). Delaying recruitment to established positions to match reduced demand or role redesign, streamlining recruitment to address increased demand and improving staff retention are common EI strategies in this classification. Examples include improving recruitment to reduce overtime, reviewing vacant roles for ongoing requirements, delaying recruitment to vacant positions where services are not required, employing medical staff to reduce locum requirements and establishing training positions to enhance recruitment.

### 5.4. Non-Clinical Business Domain

#### 5.4.1. Non-Clinical Contracts and Supplies

Although the primary purpose of healthcare services involves clinical care, a range of business functions are required to support health service operations. Asset management, asset maintenance and consumables not involved in patient care are routine considerations and as such also offer opportunities for improved efficiency. Examples of EI strategies in this category include reducing utilisation of fleet vehicles, consultants and stationery along with renegotiating commercial arrangements and changing to more cost-effective service providers.

#### 5.4.2. Finance and Operations

The finance and operations classification provides a catch-all for the range of improvements achieved through enhanced business management practices, strategic decision-making and streamlined day-to-day business activities. This classification encompasses improvements in the delivery of overall business functions which are essential for supporting the core activities and processes of delivering frontline healthcare. Examples include removing low-value equipment, reducing discretionary spending and improving routine maintenance practices.

### 5.5. Qualitative Evaluation Results

Eight responses were received from the population of 26 invitees, giving a response rate of 30.7%. A number of conclusions were able to be drawn from the survey responses as discussed below.

1.
*Please describe how you have used the NSW Health “Summary of expenses improvement strategies” resource document to support your work in efficiency improvement.*


Eight out of nine respondents addressed Question One. Responses included using the EI resource for EI ideas (two), reviewing opportunities, identifying EI strategies not currently being undertaken, promoting networking through discussion (two), supporting future planning and informing EI strategy development. Two respondents indicated that the categorisation approach was of benefit, specifically in “conducting a gap analysis to determine which categories should be investigated for further opportunities” and to “inform the category strategy development for the savings leadership program and inform our value-based procurement framework”.

2.
*Please describe your thoughts on the benefits, strengths and weaknesses of this resource document.*


Eight out of nine respondents addressed Question Two. Responses included using the EI resource for identifying EI strategy opportunities (two), for exploring and validating current EI strategies as well as for providing visibility of all EI strategies across NSW Health without having to “re-invent the wheel”. It was stated that the resource was a “very important tool to identify further opportunities”. Other respondents reported using the EI resource as a guide and comprehensive outline. Two respondents requested additional details, including implementation, financial management and process details.

3.
*Please outline your thoughts on how this resource document could be improved.*


Eight out of nine respondents addressed Question Three. Improvement suggestions included the provision of consistent high-level impact calculation methodologies, provision of hyperlinked examples for highly successful EI strategies, sharing of case studies and provision of further guidance on factors required for EI strategy success such as governance, process mapping and a culture of continuous improvement. Two suggestions related to further information on capturing, reporting and sharing EI strategy benefit measures.

4.
*What have been the impacts of the resource document on understanding and engagement with efficiency improvement within your organisation?*


Seven out of nine respondents addressed Question Four. Responses included providing EI strategy guidance to refer to (three), enabling EI discussions and supporting management framework development. The EI resource was again described as supporting NSW Health organisations to avoid “reinventing the wheel” while sharing and scaling knowledge as a “reference guide”, “providing clarity on EI plans” and “initiating positive conversations”.

### 5.6. Quantitative Observational Pre/Post Evaluation Results

The taxonomy evaluation found a 43% increase in the number of EI strategies implemented, as well as an improvement in EI category uptake in six of eight categories. This indicates that the taxonomy was associated with an increase in EI category implementation across NSW Health organisations as well as in overall volumes of EI strategies implemented. Three categories showed the most significant increase in uptake: non-clinical contracts and supplies; avoided and preventable activity; and clinical service delivery.

Results are shown in Table 2 below.

Table 3 below shows the overall number of EI strategies reported in each taxonomy category for the baseline and comparison periods.

The difference of means for pre–post changes in EI strategy numbers in each taxonomy category was 13, with a *p*-value of 0.0429, which demonstrates a statistically significant result. ANOVA or ANCOVA on gain scores were not deemed suitable due to sample size limitations.

## 6. Discussion

The complexity of identifying and evaluating the impact of EI strategies on health services is a common challenge [20]. Due to the wide array of cost pressures and potential approaches to planning improvement initiatives in public health settings, the value of creating a user guide resource to provide a point of reference has been previously identified [21,22,23]. Similarly, the utility of taxonomies in organising and mobilising knowledge for health service improvement practices has been established across a range of settings [7,8]. The taxonomy developed in this study represents a range of opportunity areas for EI which are likely to be common across public health systems internationally. This may support EI programs by serving as a guide to identify EI opportunities across these areas, as well as ensuring all areas have been addressed as part of a comprehensive EI approach. The development approach taken ensured the taxonomy and EI resource were completed and presented in a way which made them accessible and useful to the intended audience [24]. It is therefore recommended to consider aligning such resources with the terms used in practice by the intended audience. This has the potential to maximise accessibility and promote engagement should the taxonomy be applied in other settings internationally.

The findings of this study reaffirmed the breadth of various approaches to EI in public health systems. While it may therefore be expected that similar initiatives would be implemented across multiple locations within a public health system, this study demonstrated the extent to which this occurs with only one in four audited EI strategies being unique. This is indicative of the high level of duplication in EI strategies occurring across the system, thereby validating efforts aimed at reducing the amount of avoidable effort invested in such duplication. This study adds further evidence of the utility of taxonomies in promoting EI strategies within the current theoretical construct of value-based care [8]. In this context, the EI knowledge organisation and distribution supported through the taxonomy presented in this study can enable public health systems to achieve greater health service outcomes for resources invested. The taxonomy evaluation survey had a low response rate impacted by a high turnover within the study population, with only 10 of 26 potential participants remaining in their positions between taxonomy release and evaluation. While this represents a limitation to the evaluation, the survey provided feedback which supports the positive impact of sharing EI knowledge on generating EI ideas, promoting EI discussion and assisting with EI program design. This suggests that EI knowledge sharing can promote engagement with and understanding of EI. The categorisation approach was associated with supporting a systematic approach to evaluating potential for EI strategies across common areas as well as in designing local EI programs and frameworks. The EI resource was also linked to supporting and promoting EI-based discussions between broader leadership groups within NSW Health organisations. Responses indicated that the benefits of the EI resource extended beyond just the designated EI stakeholder for each NSW Health organisation, with the taxonomy described as supporting discussion of ideas, opportunities and learnings. We suggest the taxonomy has the potential to provide these positive impacts in public health systems internationally.

The EI resource was also associated with an increase in the number of NSW Health organisations implementing EI strategies across taxonomy categories. This can be linked with its benefits in supporting users to identify EI opportunities and well as promoting a comprehensive EI approach which includes EI strategies across all taxonomy categories. As the largest improvement in EI distribution by category was identified in non-clinical contracts and supplies, namely, avoided and preventable activity and clinical service delivery and patient outcomes, these categories may potentially represent areas for further investigation for additional EI strategies across all public health systems. While the implementation of the taxonomy cannot be established as the sole causal factor in these improvements, these findings are suggestive of an association between sharing knowledge of improvement approaches and an increase in the number of EI strategies developed, in addition to an increase in the number of organisations addressing opportunities across each of the taxonomy classifications.

This study highlighted the varying diversity in EI strategies across the different taxonomy categories. Two out of every five unique approaches were related to contracts and supplies, distributed close to evenly between clinical and non-clinical areas. This is likely to be reflective of the wide range of healthcare products and contracted service providers prevalent in large, complex health services. Conversely, only limited numbers of unique EI approaches focused on position creation, hiring and leave, which is potentially related to the rigid and regulated nature of industrial relations in the public health sector. The potential to scale successful EI strategies from their original setting across other similar settings within the same healthcare system is highlighted by these findings, with such efforts having the potential to result in the realisation of benefits achieved by the original initiative in similar settings. This provides new evidence for the utility of an EI taxonomy for public health systems internationally. As the policy imperative for improved efficiency continues to be reflected in the strategic priorities and performance management frameworks of public health systems, the EI taxonomy presented in this study is highly relevant to contemporary policy directions.

This study was impacted by a number of limitations. The unique setting in which the study was undertaken limited the scope and scale of investigations, while the small sample size across qualitative and quantitative evaluations limited the potential for application of complex statistical methods. The use of MS Forms limited qualitative data to free-text responses only. These limitations were overcome by clearly articulating the relevance of the study context to international settings and by using validated analysis techniques appropriate to the sample sizes.

## 7. Implications for Practice

Undertaking a systemwide audit of EI strategies can enable the mobilisation of this knowledge across a public health system, supporting EI stakeholders to address the challenge of identifying opportunities for EI initiatives and methods to measure their impact. Importantly, this has the potential to reduce repetition in the development of highly similar improvement initiatives across different areas of the system, allowing EI planners to rededicate this time to applying the shared knowledge in their individual settings to suit specific local requirements while also supporting the scaling and application of successful EI strategies [25]. This also has the potential for EI planners to use such an audit in cross-checking for gaps in their individual EI portfolios by comparing against practices in similar settings elsewhere in the system and benchmarking against peers [26,27]. Sharing evidence of successful approaches across the system can also promote engagement with EI, including promoting discussions and supporting EI program design [28]. In alignment with contemporary theoretical improvement frameworks founded on the concept of value, this study reaffirms the imperative that a focus on cost improvement is balanced with a requirement to ensure that service quality, outcomes and experiences are not adversely impacted [1]. In further alignment with established theoretical foundations in continuous quality improvement, the approach taken in this study closely follows the key steps in benchmarking methodology regarding the organisation of data to support the development of action plans [1].

The development and distribution of an EI resource provides evidence for the role of a centralised, whole-of-system approach to EI support in managing EI knowledge distributed across the system. Access to systemwide EI information, the creation of networks with EI stakeholders across the system and the ability to align EI with broader system management frameworks are key enabling benefits of a centralised EI approach. Such centralised approaches to EI are therefore well-positioned to manage the collective EI knowledge distributed across the broader system, providing structured knowledge management services and leading efforts to make this knowledge accessible and user-friendly for key EI stakeholders across the system. This can support the replication of EI success originating in a single health facility or organisation across similar settings elsewhere within public health systems internationally.

## 8. Conclusions

With the requirement to address demand and funding challenges expected to continue, public health systems will increasingly stand to benefit from centralised approaches to improving efficiency. This study finds that significant duplication of EI strategies exists across the organisations within the NSW public health system, along with significant effort unnecessarily invested in recreating EI strategies already developed elsewhere within the system. By applying audit and content management methodology as a foundational step in managing the collective body of EI knowledge in NSW Health, this study developed a categorisation system for EI strategies along with a detailed catalogue of EI strategies and their measurement approaches. The resulting taxonomy and EI resource provide vehicles for knowledge mobilisation to increase accessibility of collective EI knowledge and experience with the aim of supporting EI stakeholders to address common challenges. This study highlights the potential for centralised EI models to support public health systems to drive EI by consolidating and sharing distributed EI knowledge to address ongoing resource and demand pressures.

## Figures and Tables

**Figure 1 healthcare-11-02177-f001:**
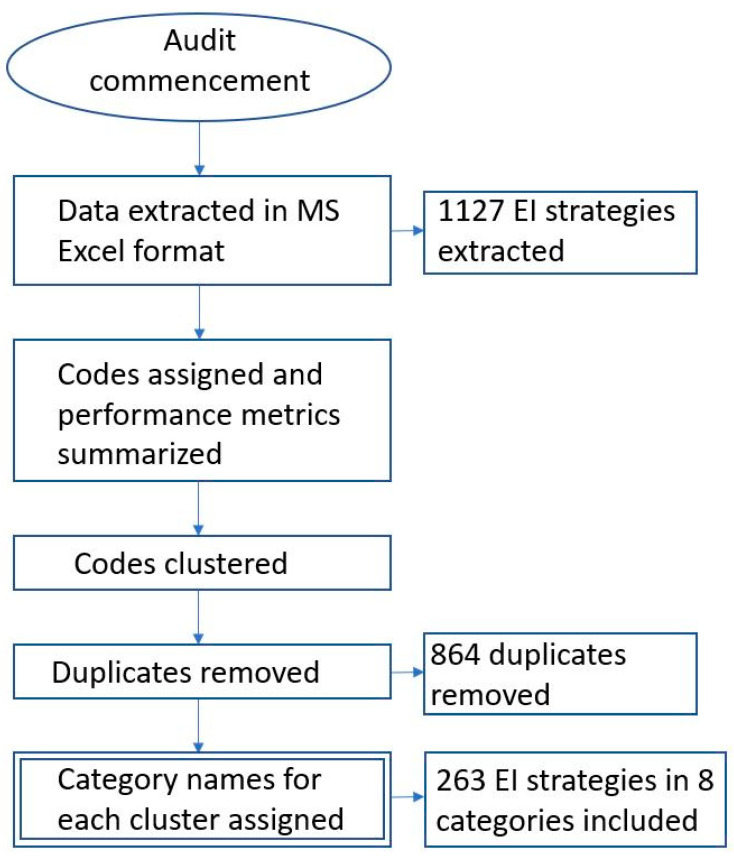
Audit and taxonomy development procedure flow chart.

**Table 1 healthcare-11-02177-t001:** Taxonomy of EI strategies.

Domain	Category	Unique EI Strategies Identified in Audit (n)	Description	Examples
Clinical Services	Clinical contracts and supplies	57	Strategies improving procurement and use of goods, consumables and commercial arrangements which are directly involved in healthcare delivery.	-Improving stock management to reduce waste. -Changing to more cost-effective service providers. -Working toward targeted reduction in high-cost consumables use.
	Clinical service delivery and patient outcomes	34	Strategies improving the processes of providing clinical care.	-Reducing unwarranted diagnostic procedures. -Improving patient transport scheduling. -Improving appointment and procedure scheduling.
Workforce	Premium staffing	35	Strategies optimising use of contingent, external and penalty-rate staffing.	-Replacing contingent staff with permanent staff. -Implementing central staffing pools. -Improving overtime governance.
	Leave management	9	Strategies ensuring appropriate and effective use of paid absence entitlements.	-Reducing excess annual leave liability. -Enhancing leave. -Reducing backfilling of positions.
	Staffing models	28	Strategies optimising mix of staff types and deployment.	-Reducing after-hours callbacks. -Maximising scope-of-practice utilisation. -Changing skill and seniority mix.
	Recruitment, vacancies and full-time equivalent (FTE)	17	Strategies related to creation and hiring of staff to positions.	-Reviewing vacant roles for ongoing requirements. -Delaying recruitment to vacant positions. -Establishing training positions to enhance recruitment.
Non-clinical business	Non-clinical contracts and supplies	59	Strategies improving procurement and use of goods, consumables and commercial arrangements which are not directly involved in the delivery of healthcare.	-Streamlining fleet vehicle services. -Reducing use of external consultants. -Changing to more cost-effective service providers.
	Finance and operations	24	Strategies improving management and administration functions.	-Removing low-value and obsolete equipment. -Reducing discretionary spending. -Improving routine maintenance practices.

**Table 2 healthcare-11-02177-t002:** Pre–post comparison of EI categories reported by NSW Health organisations, 2018–2019 and 2019–2020 financial years.

Taxonomy Category	Organisations Reporting at Least One Initiative in Category Pre-	Pre- %	Organisations Reporting at Least One Initiative in Category Post-	Post- %	Pre–Post Difference (n)	Diff %
Non-clinical contracts and supplies	11	65%	17	100%	6	35%
Avoided and preventable activity	3	18%	7	41%	4	24%
Clinical service delivery and patient outcomes	8	47%	12	71%	4	24%
Finance and operations	13	76%	15	88%	2	12%
Recruitment, vacancies and FTE	12	71%	13	76%	1	6%
Staffing models	11	65%	12	71%	1	6%
Leave management	10	59%	10	59%	0	0%
Staff engagement and development	1	6%	1	6%	0	0%
Premium staffing	12	71%	11	65%	−1	−6%
Clinical contracts and supplies	16	94%	13	76%	−3	−18%

The pre–post difference of means for the comparison of changes in EI strategy numbers by EI category were not suitable for statistical significance testing due to the small sample size.

**Table 3 healthcare-11-02177-t003:** Total improvement initiatives reported in each taxonomy category.

Taxonomy Category	Total Initiatives Pre- (n)	Total Initiatives Post- (n)	Difference (n)
Non-clinical contracts and supplies	19	70	51
Avoided and preventable activity	32	73	41
Clinical service delivery and patient outcomes	26	64	38
Finance and Operations	53	69	16
Recruitment, Vacancies and FTE	4	14	10
Staffing models	15	20	5
Leave management	37	37	0
Staff Engagement and Development	1	1	0
Premium staffing	51	42	−9
Clinical contracts and supplies	78	63	−15
Total	316	453	137

## Data Availability

Raw data relating to NSW Health EI strategies used in the development of the taxonomy, audit of EI strategies and distribution of EI strategies across NSW Health Organisations are unable to be released due to Cabinet in Confidence classification. Raw qualitative data are unable to be released as study consent did not include dissemination of raw results.

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
