# Peer review of "Efficiency Improvement Strategies for Public Health Systems: Developing and Evaluating a Taxonomy in the Australian Healthcare System"

_healthcare, 2023, doi:10.3390/healthcare11152177_

Round 1

Reviewer 1 Report

Please see attached

Author Response

Reviewer 1

The paper "Efficiency Improvement Strategies for Public Health Systems: Developing

and evaluating a taxonomy in the Australian health care system", performs a taxonomy,

defined as a catalogue of activities to be carried out by the health services in order to

develop a model of management efficiency. To this end, the authors conducted a review

of these activities implemented in New South Wales to identify areas for efficiency

improvement. After reviewing the taxonomy designed and implemented, these were

evaluated qualitatively and quantitatively.

Title and summary. The title and abstract express well the object of study, objectives,

and results of the article.

Structure of the article. The contents are well organized and they adhere to the IMRaD

structure. It includes a theoretical framework of the research problem.

Focusing on the opportunity of the study, it must be said that it is useful work since it

covers one of the major problems resulting from a health care system.

Author comments: Thank you for this positive feedback

Materials and methods.

Regarding the material and methods section, the methodology is tailored to the object of

study and the objectives and is explained in a transparent manner while it has been

validly applied to guarantee the results. However, I would like to suggest to the authors, with the intention of reinforcing the

choice of methodology, address the quantitative analysis with an ANOVA or ANCOVA

on Gain Scores.

Author comments: Thank you for this feedback. The manuscript already includes a statistical analysis using gain score (pre-post difference) averages as follows: “The difference of means for pre-post changes in EI strategy numbers in each taxonomy category was 13, with a p-value of 0.0429 which demonstrates a statistically significant result.” In our setting gain score is equivalent to the paired t-test we employed.

We have added “ANOVA or ANCOVA on gain scores were not deemed suitable due to sample size limitations”.

Results.

The results are significant and they are presented in an adequate and understandable

way not only through narration but also with self-explained tables and figures that are

also well elaborated in terms of presentation. The results justify and relate to the

objectives and methods and the results are of sufficient interest.

Discussion.

The discussion appropriately compares the study results with other works, highlighting

the main study findings.

Bibliography.

The 28.57% of the bibliography cited in the study belongs to the previous five years.

Overall, it is an interesting study and should be considered for publication in

Healthcare, once the minor revisions proposed have been resolved.

Thank you for this positive feedback.

Reviewer 2 Report

Thank you for inviting me to review this manuscript. The manuscript reports on efficiency improvement strategies for public health systems. The authors developed and evaluated a taxonomy using both qualitative and quantitative methods. Overall the manuscript reads well and merits attention. However, I think the manuscript needs minor revisions.
1. Abstract: Please add domains or categories in the result section.

2. It seems using 'strategies' for codes does not sound accurate. I recommend finding out a better expression. Strategies or alternative strategies could a few overall pictures and not so many. Please think.

Author Response

Reviewer 2:

Thank you for inviting me to review this manuscript. The manuscript reports on efficiency improvement strategies for public health systems. The authors developed and evaluated a taxonomy using both qualitative and quantitative methods. Overall the manuscript reads well and merits attention. However, I think the manuscript needs minor revisions.

  1. Abstract: Please add domains or categories in the result section.

We have added the domains to the result section of the abstract as follows:

Categories included Non-clinical contracts & supplies; Avoided & preventable activity; Clinical service delivery & patient outcomes; Finance & Operations; Recruitment, Vacancies & FTE; Staffing models; Leave management; Staff Engagement & Development; Premium staffing; Clinical contracts & supplies.

  1. It seems using 'strategies' for codes does not sound accurate. I recommend finding out a better expression. Strategies or alternative strategies could a few overall pictures and not so many. Please think.

Thank you for this feedback. The manuscript already articulates a definition for EI strategies in the context of the study as follows- “These strategies aim to reduce the cost of delivering services while also improving or maintaining quality and experience outcomes, supporting public health organisations to meet service demand within allocated budgets and maximize outcomes achieved for resources invested”. 

We suggest this is distinct from the concept of strategies more broadly and we believe this definition is sufficient to provide clarity. We have not made any amendments in relation to this feedback item.

Reviewer 3 Report

The manuscript was interesting and well-written. The following comments can help the authors to improve it.

1.       As the research has been conducted in Australia, the rational for conducting the research in that context should be explained.

2.       If the taxonomy has been evaluated qualitatively and quantitatively, please separate their findings in the result section. The results of the qualitative and quantitative studies are not clear in the current version. In addition, in the methods section, you can use subheadings such as the study design, participants and recruitment, research instrument, and data collection and analysis for each methodology used for the evaluation. The first section of the methods section can also be re-organized to improve readability.

3.       The discussion section needs to be supported by the results of similar studies.

4.       Please add the research limitations.

5.       How can other countries use the results?

6.       Please use more recent references to support your findings.

Author Response

Reviewer 3:

The manuscript was interesting and well-written. The following comments can help the authors to improve it.

  1. As the research has been conducted in Australia, the rational for conducting the research in that context should be explained.

Thank you for this feedback. We have added an explanation for this as follows:

“The presence of this centralized approach in NSW is the rationale for selecting this setting.”

Additionally, the rationale for this setting was outlined as a conflict of interest: “The corresponding author was an employee of NSW Health throughout the duration of the study”.

  1. If the taxonomy has been evaluated qualitatively and quantitatively, please separate their findings in the result section. The results of the qualitative and quantitative studies are not clear in the current version.

Results section subheadings have been updated to specify the qualitative and quantitative components.

  1. In addition, in the methods section, you can use subheadings such as the study design, participants and recruitment, research instrument, and data collection and analysis for each methodology used for the evaluation. The first section of the methods section can also be re-organized to improve readability.

Thank you for this feedback. The manuscript already uses a subheading structure for the methods including setting, design, data sources, eligibility criteria, audit procedure, taxonomy development procedure, EI resource document development implementation and evaluation, taxonomy and evaluation methodology. We note some formatting issues in the manuscript proof causing eligibility criteria, audit procedure and taxonomy development procedure subheadings to be formatted as main text may have contributed to these headings being overlooked, the formatting has been corrected.

  1. The discussion section needs to be supported by the results of similar studies.

Thank you for this feedback. The manuscript states our background review identified no similar studies for comparison, however the use of taxonomies in healthcare was cited as supporting evidence for the study design. We are therefore unable to provide comparison with results from previous studies however we have added context regarding the use of taxonomies in healthcare improvement as follows:

        “the utility of taxonomies in organizing and mobilizing knowledge for health service improvement practices has been established across a range of settings”.

  1. Please add the research limitations.

Thank you for this feedback. We have added limitations in the discussion as follows:

This study was impacted by a number of limitations. The unique setting in which the study was undertaken limited the scope and scale of investigations, while the small sample size across qualitative and quantitative evaluations limited the potential for application of complex statistical methods. The use of MS Forms limited qualitative data to free-text responses only. These limitations were overcome by clearly articulating the relevance of the study context to international settings and use of validated analysis techniques appropriate to the sample sizes.

How can other countries use the results?

This was articulated in both the discussion and implications sections in detail (lines 363-383 and lines 396-425).

  1. Please use more recent references to support your findings.

All references other than seminal works were published within ten years of the study period.

Reviewer 4 Report

Efforts should be made to enhance the theoretical and empirical significance of the research, with persuasive argument and rigorous data analysis.

Fine

Author Response

Reviewer 4:

Efforts should be made to enhance the theoretical and empirical significance of the research, with persuasive argument and rigorous data analysis.

Thank you for this feedback, however without specific guidance we are unable to action this comment. We note the differences in opinion on this matter across reviewers and concur with the number of reviewers in support of the relevance and rigour of this research.

Reviewer 5 Report

The paper reports the development, implementation, and evaluation of a novel taxonomy used in public health systems to inform system-wide EI models. Interesting paper. I have provided some additional feedback for the authors. The study is well written, but newer and current references should be provided. Please find below my comments:

1. Chapter 1 - Introduction: The Author stated that the objective of the paper is to improve the access and utilization of Efficiency Improvement knowledge at a system-wide level through the development and implementation of a taxonomy. Taxonomies can be used by researchers to find their own way to find solutions/answers to the research questions. Also, researchers are able to assess and analyze scientific studies and observations resorting to frameworks formed through taxonomies. I suggest the author identify other papers where Taxonomies are used for similar objectives.

I would like to see a deeper theoretical framework in the scope of Efficiency Improvement, specifically under the public health services. Additionally, the identified Gap needs further explanation and lacks citations. To improve the paper I would suggest the definition of research questions in order to allow a comprehensive understanding to all readers.

2. Chapter 2 - Materials and Methods: The setting section is clear and defines de boundaries of the study. There are no other studies using the same design? I suggest adding some justification why the proposed design is the most suitable for this research. 

It is quite confusing to follow all the sections under Chapter 2, too many sections that lack connection between them. As far as I can understand, the Data Source used was explored through an Archival Research method. I advise to include all the methods used. Also, if the Author is using a mixed approach (quantitative analysis and qualitative analysis) they should be stated clearly.

I understand the inductive approach, but again, I would like to see a deeper justification for this type of reasoning. 

The Sample section (line 152) has different font styles and sizes. The sampling method is confusing. I suggested rewriting this section.

Data collection section (Line 159) represents a limitation of using MS forms. I suggest including the limitations resulting from this method. 

3. Chapter 3 - Results: Lines 203, 204, and 205 are confusing, I don't understand if it is sections or text.

4. Chapter 4 and 5 - Discussion and Conclusions: The Author should try to develope the discussion around the theoritical framework (tham must be improved) and compare the results with previous studies, otherwise it looks like an audit/report paper and not a research. 

English should be reviewed at the level:

- Reduction of the length of some sentences.

Author Response

Reviewer 5:

The paper reports the development, implementation, and evaluation of a novel taxonomy used in public health systems to inform system-wide EI models. Interesting paper. I have provided some additional feedback for the authors. The study is well written, but newer and current references should be provided.

Thank you for this feedback. As stated above, all references other than seminal works were published within ten years of the study period.

  1. Chapter 1 - Introduction:The Author stated that the objective of the paper is to improve the access and utilization of Efficiency Improvement knowledge at a system-wide level through the development and implementation of a taxonomy. Taxonomies can be used by researchers to find their own way to find solutions/answers to the research questions. Also, researchers are able to assess and analyze scientific studies and observations resorting to frameworks formed through taxonomies. I suggest the author identify other papers where Taxonomies are used for similar objectives.

Thank you for this feedback. These papers were previously identified and cited in support of the selected approach (references 6, 7, 8 and 9).

I would like to see a deeper theoretical framework in the scope of Efficiency Improvement, specifically under the public health services.

Thank you for this comment. We agree in the utility of such a framework however this is beyond the scope of this study.

Additionally, the identified Gap needs further explanation and lacks citations.

The gap in evidence is comprehensively explored in our previous study which was cited as reference 1. This study was a review and synthesis of 70 studies on this topic.

 To improve the paper I would suggest the definition of research questions in order to allow a comprehensive understanding to all readers.

Thank you for this feedback. We have expanded upon the aim of the study and defined the research question as follows:

The aim of this study is to address the research question “how can access to and utilization of EI knowledge at a system-wide level can be improved through the development and implementation of a taxonomy of EI strategies?”.

  1. Chapter 2 - Materials and Methods:The setting section is clear and defines de boundaries of the study. There are no other studies using the same design? I suggest adding some justification why the proposed design is the most suitable for this research. 

It is quite confusing to follow all the sections under Chapter 2, too many sections that lack connection between them. As far as I can understand, the Data Source used was explored through an Archival Research method. I advise to include all the methods used. Also, if the Author is using a mixed approach (quantitative analysis and qualitative analysis) they should be stated clearly.

I understand the inductive approach, but again, I would like to see a deeper justification for this type of reasoning

The Sample section (line 152) has different font styles and sizes. The sampling method is confusing. I suggested rewriting this section.

Thank you for this feedback, we note this was also identified by reviewer 3 and we have amended the formatting for clarity.

Data collection section (Line 159) represents a limitation of using MS forms. I suggest including the limitations resulting from this method.

Thank you for this feedback, we have included this within the study limitations. 

  1. Chapter 3 - Results:Lines 203, 204, and 205 are confusing, I don't understand if it is sections or text.

Thank you for this feedback as also identified by other reviewers, we have corrected the formatting.

  1. Chapter 4 and 5 - Discussion and Conclusions:The Author should try to develope the discussion around the theoritical framework (tham must be improved) and compare the results with previous studies, otherwise it looks like an audit/report paper and not a research. 

Thank you for this feedback. As advised, a theoretical framework for public sector efficiency was beyond the scope of this research and no similar studies were identified in the background review for comparison.

Reviewer 6 Report

The area of the study is interesting and requires attention of scholars. The article is well written and requires only a few adjustments.

My detailed comments are as follows:

1. Lines 40-42. Please restate the objective of the study. Stating "...to improve access to and utilization of EI knowledge..." is to simplistic. Do not provide the objective of the study in the very beggining.

2. Introduction. Please explain better why the study is carried out, than present restated objective of the study. 

3. I encourage to provide more references in the introduction.

4. Discussion. Please provide more refrences, compare obtained results with the literature.

5. Consider merging the conclusion section with Implications for practice.

The style of writting might be improved, although English is of good level.

Author Response

Reviewer 6

The area of the study is interesting and requires attention of scholars. The article is well written and requires only a few adjustments.

My detailed comments are as follows:

  1. Lines 40-42. Please restate the objective of the study. Stating "...to improve access to and utilization of EI knowledge..." is to simplistic. Do not provide the objective of the study in the very beggining.

and

  1. Introduction. Please explain better why the study is carried out, than present restated objective of the study. 

Thank you for this feedback. We have specified the research question and moved the research question towards the end of the introduction rather than the beginning.

  1. I encourage to provide more references in the introduction.

and

  1. Discussion. Please provide more refrences, compare obtained results with the literature.

Thank you for this feedback. The lack of similar studies limited the reference material available for comparison.

  1. Consider merging the conclusion section with Implications for practice.

We have followed the heading guide provided in author guidelines for this submission.

Round 2

Reviewer 3 Report

I appreciate the authors for their time and efforts to revise the manuscript. It has been improved significantly.

Author Response

Thank you for this positive feedback and for your valuable guidance on opportunities to improve our manuscript. 

Reviewer 4 Report

The authors addressed the concerns.

Author Response

(The authors gave the same response as above.)

Reviewer 5 Report

Despite the changes made, in my opinion, the theoretical framework should be deeper presented. 

Author Response

Thank you for the opportunity to address further feedback on our manuscript. We have made the following amendments:

Reviewer 5:

Despite the changes made, in my opinion, the theoretical framework should be deeper presented.

We have added:

Introduction: Efficiency is conceptualised in a variety of ways in theoretical models(1). In contemporary health system paradigms, it is recognised as a central component of value-based care. In this characterisation, efficiency represents the outcomes achieved for resources invested(1,10). This exists in balance with other core elements of value, with a requirement to uphold or maintain or improve clinical, quality and experience outcomes in balance while reducing the cost of health service delivery(1).

and

Discussion: This study adds further evidence for the on the utility of taxonomies in promoting EI strategies within the current theoretical construct of value-based care(7). In this context the EI knowledge organisation and distribution supported through the taxonomy presented in this study can enable public health systems to achieve greater health service outcomes for resources invested.

and

Discussion: In alignment with contemporary theoretical improvement frameworks founded on the concept of value, this study reaffirms the imperative that a focus on cost improvement is balanced with a requirement to ensure service quality, outcomes and experiences are not adversely impacted(1). In further alignment with established theoretical foundations in continuous quality improvement, the approach taken in this study closely follows the key steps in benchmarking methodology regarding the organization of data to support the development of action plans(1).